# Direct observation of coordinated DNA movements on the nucleosome during chromatin remodelling

Anton Sabantsev[1,6], Robert F. Levendosky[2,6], Xiaowei Zhuang [3,4,5], Gregory D. Bowman[2] & Sebastian Deindl[1]

ATP-dependent chromatin remodelling enzymes (remodellers) regulate DNA accessibility in eukaryotic genomes. Many remodellers reposition (slide) nucleosomes, however, how DNA is propagated around the histone octamer during this process is unclear. Here we examine the real-time coordination of remodeller-induced DNA movements on both sides of the nucleosome using three-colour single-molecule FRET. During sliding by Chd1 and SNF2h remodellers, DNA is shifted discontinuously, with movement of entry-side DNA preceding that of exit-side DNA. The temporal delay between these movements implies a single rate-limiting step dependent on ATP binding and transient absorption or buffering of at least one base pair. High-resolution cross-linking experiments show that sliding can be achieved by buffering as few as 3 bp between entry and exit sides of the nucleosome. We propose that DNA buffering ensures nucleosome stability during ATP-dependent remodelling, and pro-vides a means for communication between remodellers acting on opposite sides of the nucleosome.

[1] Department of Cell and Molecular Biology, Science for Life Laboratory, Uppsala University, 75237 Uppsala, Sweden. [2] T.C. Jenkins Department of Biophysics, Johns Hopkins University, Baltimore, MD 21218, USA. [3] Howard Hughes Medical Institute, Harvard University, Cambridge, MA 02138, USA. [4] Department of Chemistry and Chemical Biology, Harvard University, Cambridge, MA 02138, USA. [5] Department of Physics, Harvard University, Cambridge, MA 02138, USA. [6]These authors contributed equally: Anton Sabantsev, Robert F. Levendosky. Correspondence and requests for materials should be addressed to G.D.B. (email: gdbowman@jhu.edu) or to S.D. (email: sebastian.deindl@icm.uu.se)

Eukaryotic genomes are extensively packaged into nucleosomes. With its characteristic wrapping around the histone core[1], nucleosomal DNA is largely inaccessible to most DNA-binding factors, making the nucleosome a fundamentally repressive element[2–6]. Establishment and regulation of this repressive packaging of DNA depends on ATP-dependent chromatin remodelling enzymes (remodellers), which determine occupancy, composition, and placement of nucleosomes throughout the genome[7–10]. Remodellers are essential for generating and maintaining the highly predictable and regular nucleosome organization on the vast majority of eukaryotic genes[11,12]. Central to chromatin remodelling is the ability to reposition or slide nucleosomes along DNA: sliding nucleosomes into evenly spaced arrays prevents exposure of long stretches of naked DNA, whereas sliding nucleosomes into adjacent nucleosomes stimulates eviction to create nucleosome-free regions[13–15]. Nucleosomes are essential for blocking inappropriate initiation of transcription, and the maintenance of tightly packed nucleosomes following passage of RNA polymerase II requires remodellers such as Chd1 and ISWI[16–18].

Remodellers slide nucleosomes using a superfamily 2 (SF2)-type ATPase motor[19,20] that can translocate on DNA[21,22]. For Chd1, ISWI, and SWI/SNF remodellers, DNA translocation takes place at an internal site on the nucleosome called superhelix location 2 (SHL2)[23–31]. Structural and single-molecule studies together suggest a step size of 1 nucleotide (nt) per ATP hydrolysis cycle for SF2- and SF1-type translocases[32–37]. Consistent with this expectation, a 1–2 base pair (bp) step size has been directly observed by single-molecule FRET (smFRET) for ISWI and RSC remodellers[38,39]. Interestingly, both ISWI and Chd1 remodellers have been shown to shift nucleosomal DNA in bursts of multiple bp[38,40,41].

A key question in the field is how ATP-dependent nucleosome sliding is achieved. This includes not just the initial phase of DNA translocation or a single catalytic cycle of the motor, but how local perturbations by the ATPase domain of the remodeller result in a global shift of DNA with respect to the histone octamer. Understanding how DNA moves around the histone octamer requires the observation of sliding intermediates. Such intermediates are necessarily transient, and single-molecule experiments are ideally suited to capture them. Single-molecule techniques have been utilized to visualize DNA and nucleosome translocation by remodellers[22,38–47]. Despite the important mechanistic insights gleaned from these experiments, it has remained unclear how DNA propagates around the histone core during ATP-dependent remodelling, limiting our understanding for how nucleosomes may be repositioned.

Based on indirect observations, our previous work on ISWI remodellers suggested an unexpected model of DNA movement across the histone core, where DNA first moved off the exit side of the nucleosome before new DNA was pulled onto the nucleosomal entry side[38]. Here, we present the simultaneous detection of DNA movement on both sides of the nucleosome by three-colour smFRET[48–51], which reveals a distinct order of DNA translocation. Using both Chd1 and the catalytic subunit of an ISWI-type remodeler, SNF2h, we show a clear delay between translocation of DNA onto and off of the nucleosome, with DNA first shifting onto the entry side. After an initial round of ATP hydrolysis and movement of entry-side DNA, at least one additional ATP-binding event occurs before DNA movement is propagated all the way to the exit side, suggesting that the nucleosome can absorb one or more bp of translocated DNA between the ATPase binding site and the exit side of the nucleosome. Using site-specific cross-linking, we observe that Chd1 can shift the DNA to the exit side by 4 bp even in the presence of a gap that blocks ATPase translocation after 4–5 bp.

Taken together, these results suggest that nucleosomes transiently absorb 1–3 bp during ATP-dependent sliding. We propose that this DNA-buffering capability of the nucleosome provides a potential means for communication and regulation of remodellers bound to opposite sides of the nucleosome, ensuring nucleosome stability during chromatin remodelling.

## Results

**Simultaneous observation of entry- and exit-side DNA movement.** FRET is well suited for probing how chromatin remodellers dynamically alter nucleosome positioning[38–41,52]. A common approach has been to monitor one side of the nucleosome as DNA moves either onto or off of the histone core, using FRET between a single donor-acceptor pair. While extremely informative about local changes, a single FRET pair cannot resolve how DNA movements on opposite sides of the nucleosome are coordinated. Chromatin remodellers translocate on DNA from a fixed position on the nucleosome, which results in a global shift of DNA around the histone octamer. To study coordination of this process, we monitored both nucleosomal entry and exit sides simultaneously by three-colour smFRET[48–51].

We placed two different dyes (Cy3 and Alexa750) on the DNA on opposite sides of the nucleosome, as well as a centrally positioned dye (Cy5) on the histone core (Fig. 1a and Supplementary Fig. 1). The central dye serves as an acceptor for one DNA dye and a donor for the other, thus generating two overlapping FRET pairs. Given the inherent two-fold symmetry of the nucleosome with two copies of each histone, one challenge is obtaining a homogenous population of nucleosomes with a single label on one side of the nucleosome disk. Heterogeneity in nucleosome labelling was previously observed and resolved into distinct populations in two-colour FRET experiments;[38,41] however, such heterogeneity would be prohibitive for more complex three-colour FRET experiments. We therefore exploited the asymmetry of the Widom 601 positioning sequence[53] to generate asymmetric nucleosomes from hexasomes[54]. By starting with 601 hexasomes, each H2A/H2B dimer can be deposited separately, allowing for a homogenous population of nucleosomes with a single, uniquely positioned Cy5-labelled H2A/H2B dimer (Fig. 1a).

Chd1 is known to shift nucleosomes away from short DNA ends[26,55]. Here we generated end-positioned nucleosomes, and therefore refer to the side with short DNA as the exit side, and the side with a long (79 bp) extranucleosomal segment of DNA as the entry side (Fig. 1a; Supplementary Table 1). Nucleosomes were immobilized onto a PEG-coated quartz surface and fluorescence signals from individual nucleosomes were monitored using a total-internal-reflection fluorescence (TIRF) geometry (Fig. 1b). In order to unambiguously determine the FRET contributions of the three fluorophores, we used alternating excitation with 532 and 638 nm lasers to excite Cy3 and Cy5, respectively, and calculated entry- and exit-side FRET values (see Methods).

Our initial design (Supplementary Fig. 1) adopted locations of FRET dyes previously utilized to observe nucleosome translocation in two-colour bulk and single-molecule experiments, where the DNA labels are located just outside where DNA wraps around the histone core. This design, however, proved impractical for studying remodelling by Chd1. Nucleosomes labelled in this manner exhibited rapid fluctuations in FRET from Cy3 upon addition of Chd1 without ATP (Supplementary Fig. 1b), most likely due to previously observed Chd1-induced DNA unwrapping from the nucleosome[28,30,56,57]. To selectively monitor DNA translocation without interference from unwrapping, we devised a different design (Fig. 1a), with both entry- and exit-side fluorophores 18 and 24 bp inside the edge of the nucleosome,

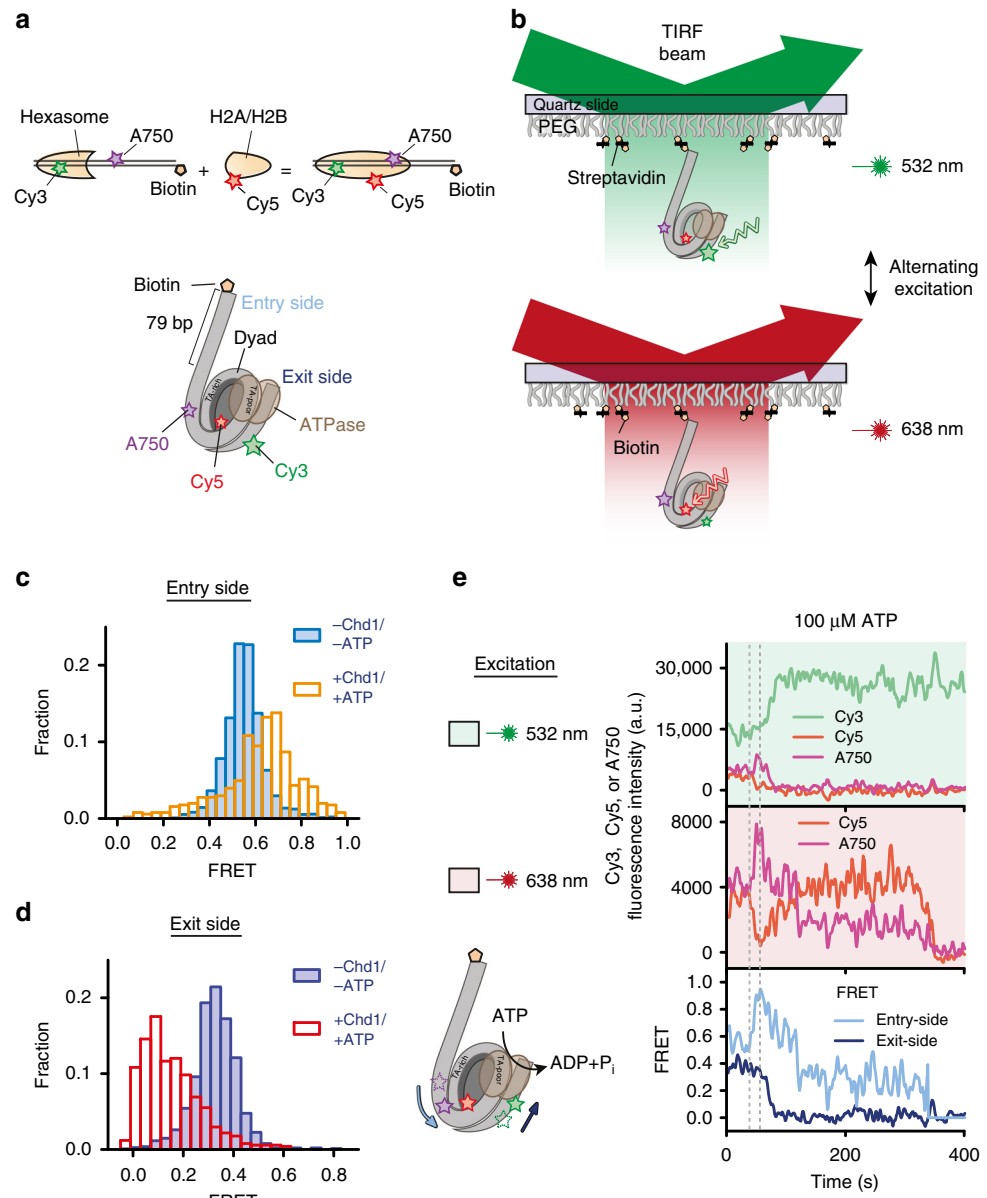

**Fig. 1** Simultaneous detection of entry- and exit-side movements by three-colour FRET. **a** Cartoon representations of the three-colour labelling scheme. The ATPase motor of the remodeller is shown in brown and Cy3, Cy5, and Alexa750 are depicted by green, red, and purple stars, respectively. The entry side is on the TA-poor side of the 601 sequence, and the exit side is on the TA-rich side. **b** Schematic of FRET detection with alternating 532 nm and 638 nm laser excitation of nucleosomes labelled with Cy3, Cy5, and Alexa750. Entry-side (**c**) and exit-side (**d**) FRET histograms constructed from 1543 traces before (solid bars) and 507 traces after (open bars) remodelling by 300 nM Chd1 in the presence of 1 mM ATP. **e** Representative Cy3 (green), Cy5 (red), and Alexa750 (purple) fluorescence and FRET time traces (entry-side, light blue; exit-side, dark blue) showing sliding of a single nucleosome after addition of 300 nM Chd1 and 1 mM ATP. Time traces were recorded with alternating 532 nm and 638 nm laser excitation, as indicated by the areas shaded in green and red, respectively. The dashed lines indicate the onset of changes in entry-side and exit-side FRET. Entry- and exit-side FRET values, determined simultaneously in the same experiment using alternating laser excitation, resembled those separately determined using corresponding two-colour FRET nucleosomes possessing Cy5-H2B and either an exit- or entry-side DNA fluorophore, respectively (Supplementary Fig. 2). Source data are provided as a Source Data file

respectively, and the Cy5 dye on the H2B C-terminus (Fig. 1a; Supplementary Table 1).

Using alternating laser excitation (Fig. 1b), we calculated entry-side FRET values as FRET originating from Cy5, and exit-side FRET values as FRET originating from Cy3. The presence of only a single Cy5-labelled H2B on each histone octamer, as manifested in a single photobleaching step (Supplementary Fig. 3a), gave rise to single entry-side as well as exit-side FRET populations (Supplementary Fig. 4). Both entry- and exit-side FRET

distributions were unchanged when Chd1 was added in the absence of ATP or with ATP-γ-S, indicating that the internal three-colour configuration was not affected by Chd1-induced DNA unwrapping (Supplementary Fig. 4a). As shown by native gel sliding experiments, Chd1 was capable of repositioning these labelled nucleosomes (Supplementary Fig. 3b). When Chd1 and ATP were added to surface-immobilized nucleosomes, both entry- and exit-side FRET exhibited marked changes (Figs. 1c–e) that were consistent with repositioning of the histone octamer

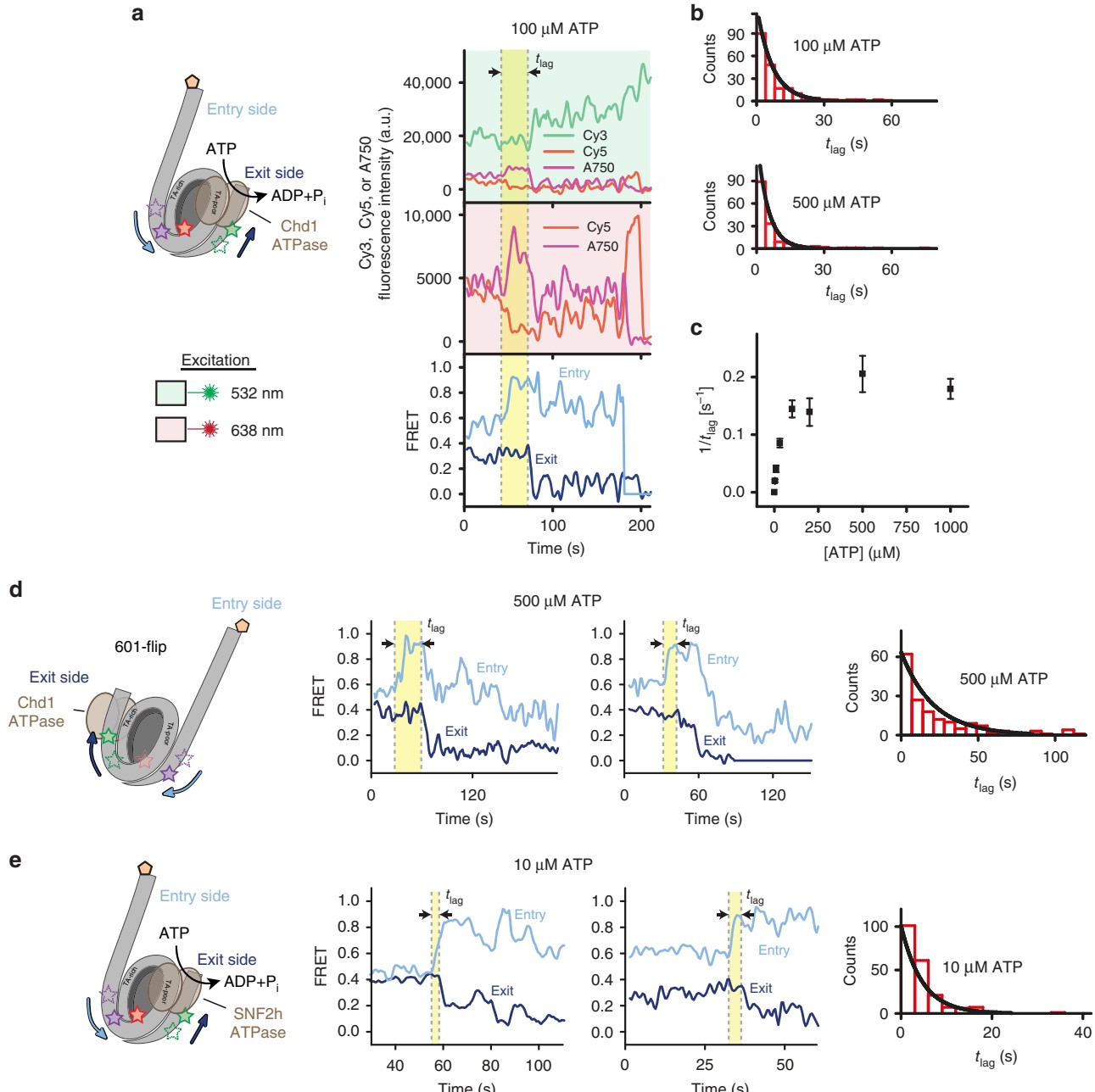

**Fig. 2** Entry-side movement of nucleosomal DNA precedes exit-side movement. **a** Cy3 (green), Cy5 (red), and Alexa750 (purple) fluorescence and FRET time traces (exit-side: dark blue; entry-side: light blue) showing sliding of a single nucleosome, after addition of 300 nM Chd1 and 100 μM ATP. The lag time between the onset of entry- and exit-side DNA movements is denoted as $t_{lag}$. Time traces were recorded with alternating 532 nm and 638 nm laser excitation, as indicated by the areas shaded in green and red, respectively. **b** Histograms of $t_{lag}$ values (red bars) observed with 100 or 500 μM ATP, fit with a single-exponential distribution ($N = 185$–220 events). **c** Dependence of $1/t_{lag}$ on the ATP concentration. Data are shown as the mean ± SEM ($N = 40$–340 events). **d** Left: Cartoon schematic showing the 601-flip nucleosome, with short exit DNA on the TA-poor side and long entry DNA on the TA-rich side of the 601 sequence. Middle: Two representative FRET time traces (exit-side: dark blue; entry-side: light blue) showing sliding of single 601-flip nucleosomes after addition of 300 nM Chd1 and 500 μM ATP. Right: Histogram of $t_{lag}$ values (red bars) from many 601-flip nucleosomes observed with 500 μM ATP, fit with a single-exponential distribution ($N = 190$ events). **e** Nucleosome sliding by SNF2h shows the same coordination of entry/exit DNA movements. Middle: Two representative FRET time traces (exit-side: dark blue; entry-side: light blue) showing sliding of single nucleosomes after addition of 1 μM SNF2h and 10 μM ATP. Right: Histogram of $t_{lag}$ values (red bars) constructed from 226 events, fit to a single-exponential distribution (black line). Source data are provided as a Source Data file

onto the long entry-side DNA and away from the short exit side. After an initial waiting period, remodelling time traces exhibited an increase in entry-side FRET, signifying that the Alexa750 fluorophore moved further onto the nucleosome (Fig. 1e). As DNA continues to move, eventually the Alexa750 dye is expected

to pass the Cy5 dye, leading to a FRET decrease. Indeed, we recorded such a non-monotonic change of entry-side FRET. On the exit-side, FRET decreased upon ATP-dependent action of Chd1, indicating that Cy3 moves away from Cy5 as DNA shifts toward the exit side of the nucleosome (Fig. 1e).

**DNA moves discontinuously during nucleosome sliding**. Our ability to simultaneously monitor both entry and exit sides of the nucleosome enables the real-time investigation of how remodeller-induced DNA movements may be coordinated. Strikingly, upon addition of Chd1 and ATP, time traces consistently exhibited changes in entry-side FRET before any exit-side FRET changes occurred (Fig. 2a). Such coordination indicates discontinuous movement of nucleosomal DNA, with the entry side preceding the exit side. The delay in exit-side movement further implies that during remodelling, the nucleosome accommodates an additional one or more bp of DNA between entry-side and exit-side fluorophores, likely between the entry SHL2 site, where the ATPase translocates DNA, and the exit-side fluorophore.

**Exit-side movement involves an additional ATP-binding event**. For Chd1, the $K_M$ for ATP was previously determined to be 50–60 μM[58]. At subsaturating ATP concentrations, nucleotide binding limits the observed rate of nucleosome sliding[38,41,47]. We wondered whether varying ATP concentration would also influence the coordination between entry-side and exit-side DNA movements. We therefore determined the lag time, $t_{lag}$, between the onset of FRET changes at entry and exit sides of the nucleosome at various ATP concentrations (Figs. 2a–c). With increasing ATP concentrations, the delay between entry-side and exit-side DNA movements decreased (Figs. 2b, c), indicating that entry-exit coordination is sensitive to action of the ATPase.

The 601 nucleosome positioning sequence is notably asymmetric, which has been shown to affect both unwrapping characteristics and the preferred direction of sliding by Chd1[59,60]. A characteristic feature of the 601 is a set of periodic TpA (TA) dinucleotide steps that is more prevalent on one side of the dyad[61], referred to as the TA-rich side. In our nucleosome construct, the long entry-side linker was adjacent to the TA-poor side of the 601 sequence, and thus the exit-side DNA corresponded to the TA-rich side (Fig. 2a). To examine whether the observed entry-then-exit order of DNA movements might be affected by the orientation of the 601 sequence, we constructed 601-flip nucleosomes, which had the long DNA linker (entry side) on the TA-rich side of the 601 and the exit on the TA-poor side (Fig. 2d). Notably, the 601-flip nucleosomes displayed the same order of events during remodelling, with movement on the entry-side of the nucleosome preceding movement on the exit side, albeit with an overall longer lag time (at 500 μM ATP, $t_{lag} = 4.9 \pm 0.8$ s ($N = 185$ events) versus $t_{lag} = 21.0 \pm 2.2$ s ($N = 190$ events) for the original and flipped orientations, respectively). Thus, DNA sequence alters the magnitude of the entry-exit delay but the overall order of the sequential DNA movements is independent of the orientation of the 601 sequence.

**A similar entry-then-exit coordination of DNA by SNF2h**. Like Chd1, ISWI-type remodelers also slide nucleosomes by acting at SHL2 (refs. [24,25]). ISWI-type remodelers differ from Chd1 by possessing one or more auxiliary subunits that accompany the catalytic, ATPase-containing subunit[7]. For example, ACF, a human ISWI remodeller consists of a catalytic subunit SNF2h and an accessory subunit ACF1. To see if both Chd1 and the ISWI catalytic subunit promote a similar coordination of entry- and exit-side DNA movements, we tested the SNF2h catalytic subunit with our three-colour FRET setup (Fig. 2e). Similar to Chd1, nucleosome sliding by SNF2h consistently produced FRET changes on the entry side prior to those on the exit side, also with a single-exponential lag-time distribution (Fig. 2e and Supplementary Fig 4b). These results suggest that nucleosomes respond similarly to Chd1 and SNF2h, where the DNA is first pulled onto

the entry side of the nucleosome before shifting off the exit side. Previous work, using a yeast ISWI enzyme consisting of a catalytic subunit (Isw2, homologous to SNF2h) and three accessory subunits (Itc1, Dpb4, and Dls1), suggested a different timing of DNA movements at the entry and exit sides[38]. While this discrepancy may possibly be due to the absence of accessory ISWI subunits in the current experiment, a likely explanation is the asymmetry of the 601 positioning sequence. The lag between remodeler binding and DNA movement is longer when the TA-rich side of the 601 sequence is on the entry side of the nucleosome as compared to the flipped orientation (Figs. 2c, d). In the previous experiment, timing was in part inferred from two separate two-colour FRET experiments, where the TA-rich portion of the 601 sequence was either on the entering or the exiting half of the nucleosome[38]. Here, by simultaneously observing both sides of the same nucleosome using three-colour FRET, we more directly probe the coordination. For all ATP concentrations tested, the histograms of observed lag times were well-described by single-exponential distributions (Figs. 2b, d, e), indicative of a single, rate-limiting event. The dependence of lag time on ATP concentration suggests that this single, rate-limiting event involves ATP binding. Thus, the onset of exit-side movement is likely triggered by binding and possibly hydrolysis of an additional ATP by the remodeller.

**DNA translocation by 4–5 bp can reposition the nucleosome**. Previously, we observed that upon binding, the ATPase of Chd1 pulls DNA onto the nucleosome by 1–2 bp, forming a twist defect on the entry side SHL2 site[62]. The twist defect only provides temporary storage for one additional bp, however, since binding and hydrolysis of ATP destabilizes the twist defect at the entry SHL2 and effectively shuttles DNA toward the dyad[62]. Here, the ATP-dependent lag times we observed (Fig. 2c) suggest that one or more additional ATP-binding events occur before DNA propagation reaches the exit side of the nucleosome. These observations are consistent with a sliding mechanism where the nucleosome transiently absorbs one or more additional bp of DNA.

To further examine this unexpected absorption of DNA on the nucleosome, we constructed nucleosomes with 2-nt single-stranded DNA (ssDNA) gaps near the entry SHL2, to limit the extent of DNA translocation. Gaps at SHL2 do not disrupt the wrapping of DNA around the histone core, yet have been shown to block nucleosome sliding by Chd1 and other remodelers[23–27]. Using end-labelled DNA, single-molecule experiments further showed that gaps can be used to allow limited translocation of nucleosomal DNA, with the observation that nucleosomal DNA can shift until the gap reaches ~21–22 bp from the dyad[38,41]. Thus, a gap located $m$ bp from this 21–22 bp site should allow for an $m$ bp shift of nucleosomal DNA.

Since Chd1 and SNF2h shift DNA on the entry side prior to exit-side movement, we wondered how much DNA is required to shift onto the entry side before DNA also starts shifting off of the exit side. To address this question, we first sought to probe the remodelling of gapped, three-colour nucleosomes that would allow limited movement of entry DNA. Surprisingly and in stark contrast with ungapped nucleosomes, nucleosomes with 2 nt gaps at $m = 0$, $m = 3$, $m = 5$, and $m = 8$ bp displayed changes in FRET to varying extents upon addition of Chd1 alone or with ATP-γ-S (Supplementary Fig. 5). While the internal dyes of three-colour nucleosomes allow for ATP-dependent nucleosome sliding, the combination of the internal dyes plus a 2 nt DNA gap at the entry SHL2 may give rise to structural perturbations of DNA that produce FRET changes independently of ATP hydrolysis. Further, FRET is sensitive to transient, non-canonical placements

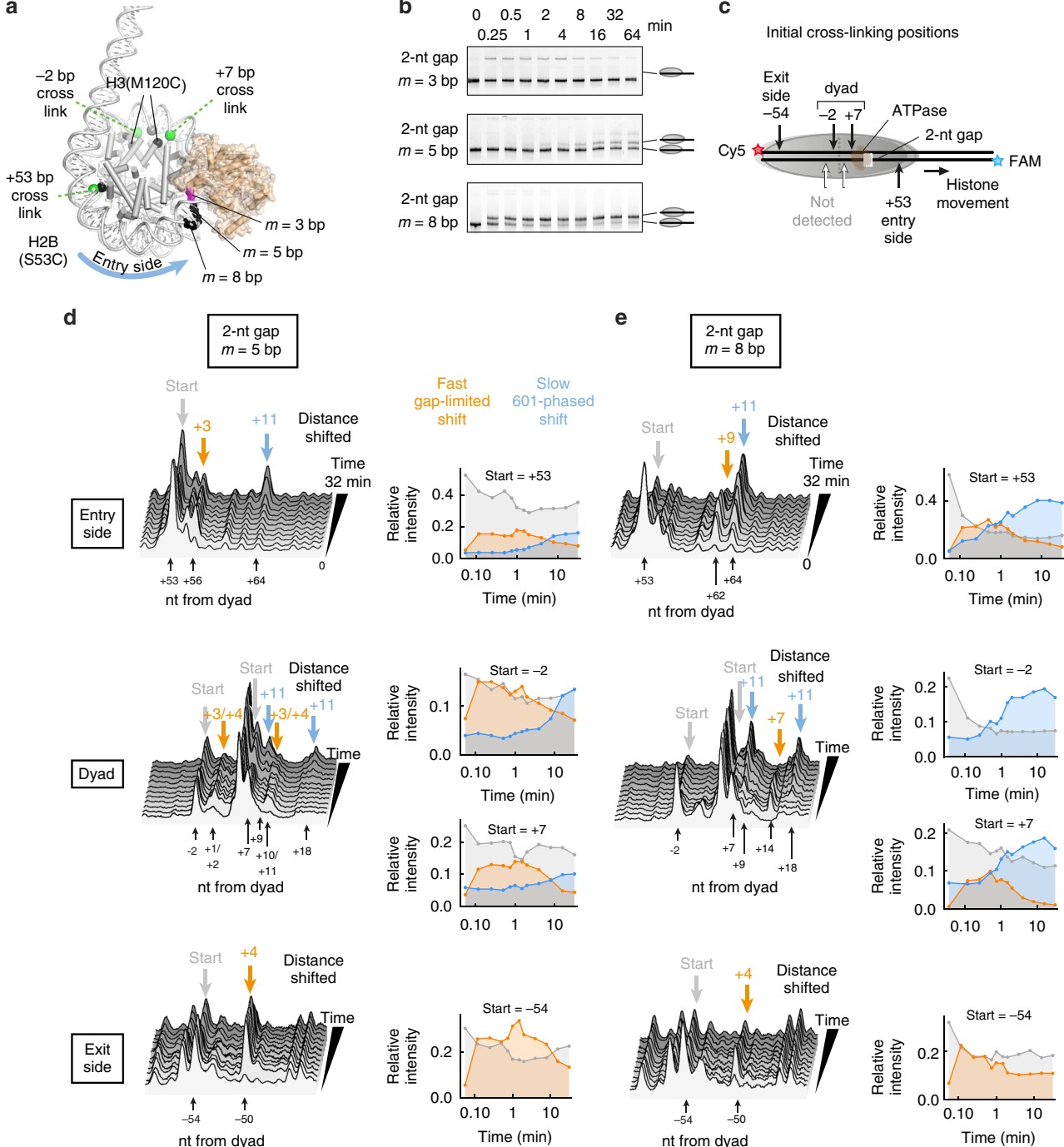

**Fig. 3** Buffering 1 to 3 bp is sufficient for allowing nucleosome repositioning. **a** Model of the nucleosome-Chd1 structure (pdb code: 5O9G [ref. [28].]) showing the position of the ATPase motor (brown) relative to 2 nt ssDNA gaps (black) that limit nucleosome sliding. **b** Native gel electrophoretic mobility assay comparing sliding of nucleosomes with ssDNA gaps at $m = 3$, $m = 5$, or $m = 8$ nt from the entry-side SHL2. Reactions contained 150 nM nucleosome, 200 nM Chd1 and 1 mM ATP. The supershifted bands at the top of each gel represent Chd1-nucleosome complexes. Gels are representative of two or more independent replicates. **c** Schematic depicting site-specific cross-linking positions on the nucleosome. Using nucleosomes containing two cysteine substitutions, three regions were simultaneously followed. Nucleosome sliding by Chd1, which shifts the histone core toward the side with longer linker DNA (to the right), can be observed by corresponding shifts of DNA cross-linking sites. **d**, **e** Time-course experiments showing Chd1-dependent changes in DNA positioning of nucleosomes containing gaps at $m = 5$ bp (**d**) and $m = 8$ bp (**e**). At each cross-linking site, stacked intensity plots showing time-dependent changes in the distribution of cross-links are given on the left, and the relative intensities of starting versus shifted cross-linking products are plotted on the right. Cross-linking products that rapidly appeared and differed depending on the location of the gap are indicated in orange. Products that formed more slowly over time, which correspond to the preferred phase of the 601 sequence, are shown in blue. Note that consistent with previous work[54], cross-linking on the exit side did not show a nucleosome population shifted by +11 bp (blue), likely due to sequence bias in cross-linking. Reactions contained 150 nM nucleosome, 200 nM Chd1, and 1 mM ATP. The time-course for these experiments was 0″, 7″, 15″, 30″, 45″, 60″, 90″, 2′, 4′, 8′, 16′, and 32′. Results are consistent across three separate experiments, two of which use similar time courses. Gel images are shown in Supplementary Fig. 6. Source data are provided as a Source Data file

of the DNA on the nucleosome, which could obscure detection of ATP-dependent DNA translocation.

We therefore turned to site-specific histone-DNA cross-linking to follow limited DNA movements around the histone core using gapped nucleosomes (Fig. 3). With this technique, single cysteines are labelled with a photo-reactive cross-linker, which can produce nicks in the DNA backbone at sites of cross-linking[62–64]. Shifts in cross-linking positions, monitored on urea denaturing gels, are interpreted as relative displacements in DNA past the cross-linking site on the histone core. To detect DNA movement at several sites simultaneously, we produced nucleosomes with two distinct cysteines: H2B(S53C), which cross-links on the outer gyre of DNA and therefore reports on both the entry and exit side, and H3(M120C), which forms dual cross-links around the nucleosome dyad (Figs. 3a, c).

To limit nucleosome sliding by Chd1, we tested nucleosomes with gaps at $m = 3$, 5, or 8 bp. By native gel, the three gapped nucleosomes responded differently upon addition of Chd1 and ATP. The $m = 3$ bp gapped nucleosome showed no changes in mobility, and while both $m = 5$ and $m = 8$ bp gapped nucleosomes showed similar band shifts, the $m = 5$ bp construct was repositioned much more slowly than $m = 8$ bp (Fig. 3b). To examine these sliding products at a resolution of 1–2 bp, we labelled the nucleosomes with the photo-cross-linker azidophenacyl bromide and performed UV cross-linking in the presence of Chd1. In parallel to the native gel experiments, cross-linking showed robust sliding for $m = 8$ bp gapped nucleosome, a slower and weaker response for $m = 5$ bp gapped nucleosome, and no apparent shifts for the $m = 3$ bp gapped nucleosome (Supplementary Fig. 6). The Chd1-dependent shifts in histone cross-links for the $m = 5$ bp and $m = 8$ bp gapped nucleosomes were dependent on ATP hydrolysis, as reactions without nucleotide or with ATP-γ-S did not alter the cross-linking pattern (Supplementary Fig. 7)

For cross-linking reactions with $m = 5$ and $m = 8$ bp gapped nucleosomes, the shifts in cross-linked species over time are shown as intensity plots in Fig. 3d, e. Both of these gapped nucleosomes exhibited fast DNA movement corresponding to a short distance shifted (orange) induced by the remodeler before interference from the ssDNA gap, followed by a slower and farther movement corresponding to an ~11-bp shift (blue) (Figs. 3d, e). Since an 11-bp shift corresponds to a full helical turn of DNA, the slower shifts recovered the preferred phasing of the 601 positioning sequence and were likely due to nucleosome relaxation. Consequently, time traces exhibited a transient accumulation of gap-limited cross-links (orange) whose subsequent decrease was accompanied by an increase in 601-phased cross-links (blue).

For the $m = 5$ bp nucleosomes (Fig. 3d), the fast gap-limited shifts corresponded to movements of 3–4 nt. These were observed not only at the entry side (top) and dyad (middle), but also at the exit side (bottom), indicating that 4 bp of translocation by the ATPase were sufficient to propagate DNA all the way to the exit side of the nucleosome. For the $m = 8$ bp nucleosomes (Fig. 3e), gap-limited shifts were observed that corresponded to movements of 9, 7, and 4 nt at entry side (top), dyad (middle), and exit side (bottom), respectively. We note that in previous studies, we found sequence bias in cross-linking to the 601, where no H2B(S53C) cross-links were detected with the histone octamer shifted off the TA-rich side of the 601 by 10–11 bp[54]. This sequence bias likely obscures the exit-side cross-linking products corresponding to shifts beyond 4 nt. In agreement with this, intensities corresponding to 4-nt exit-side cross-links eventually decreased for both gapped nucleosomes (see intensity plots in Figs. 3d, e, bottom). Although gap-limited shifts (orange) accumulate rapidly in both cases (Figs. 3d, e), their subsequent decrease and the concomitant

increase in 601-phased cross-links (blue) occurred substantially more rapidly for $m = 8$ bp nucleosomes. The gap at $m = 5$ bp therefore presented a stronger barrier, making it more difficult for the remodeller to successfully maintain DNA shifted off the 601 positioning sequence by a few bp.

The lack of changes in cross-linking for the $m = 3$ bp gapped nucleosome (Supplementary Fig. 6) may be due to the close proximity of the gap with the ATPase binding site. To examine how gaps may affect Chd1-nucleosome interactions at SHL2, cross-linking was carried out with a Chd1(N650C) variant. Interestingly, the $m = 3$ bp gapped nucleosomes cross-linked to the Chd1 ATPase motor more strongly than the other substrates (Supplementary Fig. 8), which may reflect subtle changes at the binding site due to the gap. Despite the inability of Chd1 to reposition the $m = 3$ bp gapped nucleosome, the observed shifts for $m = 5$ bp and $m = 8$ bp gapped nucleosomes suggest that the nucleosome need not absorb more than 3–4 bp during the Chd1 remodelling reaction. Therefore, Chd1-catalyzed remodelling can occur with as little as 1–3 bp absorbed on the nucleosome.

## Discussion

A key unanswered question in the remodelling field is how movements of nucleosomal DNA around the histone core are coordinated during ATP-dependent nucleosome sliding. In this work, we used three-colour FRET imaging to simultaneously monitor entry and exit sides of individual nucleosomes, directly observing the relative timing of DNA movements at different sites on the nucleosome during remodelling.

Our results suggest that during ATP-dependent translocation at SHL2, segments of DNA are moved discontinuously on the nucleosome. Based on the orientation and position of the Chd1 ATPase motor on the nucleosome, DNA is expected to be shifted from the entry side SHL2 toward the dyad[28,30,31]. Our three-colour FRET experiments showed that DNA movement initially occurs on the entry side, powered by ATP hydrolysis, yet does not immediately reach the exit side. Instead, our data for Chd1 indicate that the nucleosome can transiently accommodate one or more base pairs of extra DNA before a front of DNA movement reaches the exit side.

For Chd1, the temporal delay between DNA movements monitored at entry and exit sides of the nucleosome was more pronounced at limiting ATP concentrations. At each ATP concentration, the distributions of lag times fit to single exponentials, indicating a single rate-limiting step prior to exit-side movement. Although nucleosomes made with the 601 positioning sequence are known to have asymmetric properties[5,54,59], we observed the same order of entry-side movement followed by exit-side movement with a flipped orientation of the 601 (Fig. 2d). For the flipped 601, the delay prior to exit-side movement was longer yet also produced a single-exponential distribution of $t_{lag}$. Thus, while DNA sequence can affect the magnitude of the time delay, the coordination that determined the sequence of entry and exit DNA movement was not dependent on 601 orientation.

How much DNA can the nucleosome absorb or buffer during ATP-dependent sliding? Histone mapping experiments with gapped nucleosomes showed that limiting the ability of the Chd1 ATPase to translocate DNA by ~4–5 bp at the entry-side SHL2 still allowed for a 4-bp shift on the exit side of the nucleosome (Fig. 3). Our single-molecule FRET and cross-linking experiments suggest therefore that 1–3 bp can be absorbed between the entry and exit side, likely between the entry SHL2 site and exit side during Chd1 remodelling. While we cannot rule out the possibility that nucleosomes may transiently accommodate more DNA, our data suggest that the Chd1 remodeller need not form large loops to reposition nucleosomes.

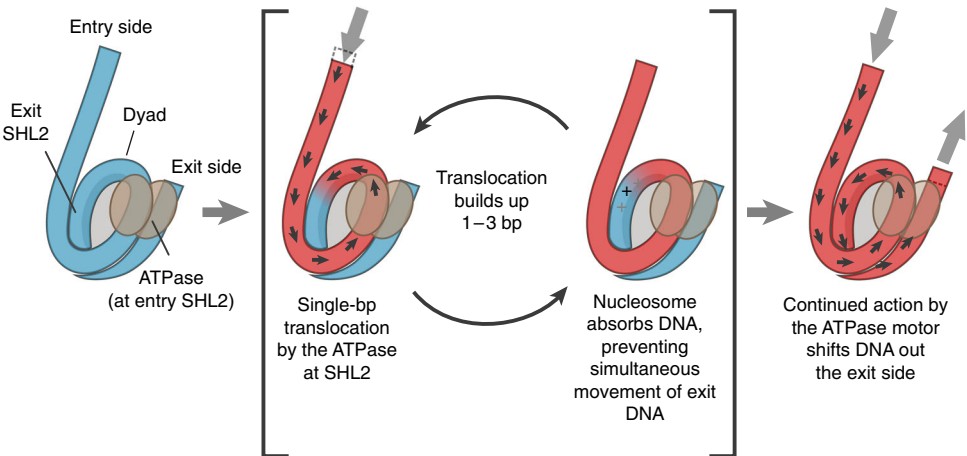

**Fig. 4** A model for nucleosome sliding by Chd1 and ISWI. The remodeller ATPase (brown), located at SHL2, initiates nucleosome sliding on un-shifted DNA (blue). Cycles of hydrolysis translocate DNA at the SHL2 site, drawing DNA onto the entry side of the nucleosome and pushing DNA towards the exit side in ~1 bp steps (shifted DNA shown in red). While some segments of nucleosomal DNA immediately twist to convey the additional bp, other segments absorb the translocated DNA, resulting in local twist defects (+) on the nucleosome. Creation of these twist defects delays the movement of DNA to the exit side

We propose that remodelling by Chd1 and SNF2h, and possibly ISWI-family remodelers in general, is achieved through a twist diffusion mechanism, where the ability of the nucleosome to absorb DNA is responsible for discontinuous movement of DNA around the histone core (Fig. 4). We find that the buffering capability of the nucleosome allows for initial translocation of DNA on the entry side, without simultaneous DNA movement on the exit side. Recent work has indicated that, in coordination with the ATP binding and hydrolysis cycle, Chd1 stimulates formation and then elimination of twist defects at the entry SHL2 (ref. [62]). Based on results we describe here, we propose that the observed lag in exit-side movement arises from one or more twist defects occurring between the entry SHL2 and the exit side (Fig. 4). While the present experiments cannot reveal the precise location of twist defects, variation in DNA length has been observed in several nucleosome crystal structures at SHL2 as well as SHL5 (refs. [1,65–67]), and recent molecular simulations have suggested that SHL1 may accommodate an additional bp more easily than neighbouring DNA segments[68,69]. In accordance with the twist diffusion model[70–72], translocation of DNA initiated at the entry SHL2 would be absorbed at one or more sites, thereby introducing twist defects.

We envision two possible scenarios whereby continued translocation by the ATPase motor at the entry-side SHL2 successfully propagates DNA out the exit side (Fig. 4). Twist defects arise when DNA duplex geometry locally changes to accommodate an additional bp, thereby interrupting the corkscrew motion of DNA from being transferred to the neighboring segment. In one scenario, once a twist defect has reached its buffering capacity, torque applied to the upstream DNA segment would be immediately transferred to the downstream segment. A nucleosome may have one or more sites where twist defects can be accommodated, and a saturating number of twist defects would enable translocation at the entry SHL2 to simultaneously shift DNA out of the exit side of the nucleosome. As twist defects accumulate, it is likely that the ATPase motor would have to overcome increasing energetic barriers to translocate DNA until DNA starts shifting out of the exit side. The last translocation event that initiates this movement would exhibit the highest barrier, which could explain the observed single ATP-dependent rate-limiting step in $t_{lag}$.

In another scenario, the transfer of DNA from the entry SHL2 site to the exit side may result from the spontaneous resolution of twist defects. The delay between entry- and exit-side movements could therefore simply reflect the time it takes for a twist defect to collapse, triggering rotation/translation of the downstream DNA segment, and eventually propagating the additional bp of DNA to the exit side. As each twist defect is expected to add strain to the nucleosomal DNA, an accumulation of multiple twist defects could lower the energy barrier for twist defect collapse. Thus, the rate at which twist defects are produced, through action of the ATPase motor at the entry SHL2, would correlate with the rate of DNA translocation out the exit side of the nucleosome, giving rise to an ATP-dependent $t_{lag}$.

Although this work focuses on Chd1 and SNF2h, we expect that the intrinsic DNA-buffering ability of the nucleosome observed here is a universal property that all remodelers must accommodate when altering nucleosome positioning. Interestingly, sliding by both Chd1 and ISWI-family enzymes has been reported to occur in multi-bp bursts[38,40,41]. While such bursts of 1-bp translocation steps may stem from remodeller-specific regulation of the ATPase motor, such behaviour may also reflect the DNA-buffering ability of the nucleosome.

What are the implications of the observed DNA absorption by the nucleosome during ATP-dependent sliding? One intriguing possibility is that DNA buffering could provide a means for remodeller-remodeller communication on the nucleosome. For some classes of remodelers such as Chd1 and ISWI, two remodeller molecules can simultaneously bind to a nucleosome, one at each of the two SHL2 sites on opposite sides of the dyad[30,31,73]. With this symmetric arrangement, the SHL2 site where one remodeller ATPase is bound corresponds to the exit-side SHL2 of the other remodeller. With SHL1 and SHL2 sites serving as potential reservoirs for twist defects[68,69], the buffering capability of the nucleosome suggests that the twist defect created by one remodeller during DNA translocation could transiently affect the ATPase binding site of the opposing remodeller. We speculate that twist defects created by one ATPase could therefore interfere with translocation by the other, providing a simple mechanism for coordinating action of remodelers on opposite sides of the nucleosome. One important consequence of such coordination would be preventing simultaneous action of two remodelers bound at each of the two SHL2 locations, which would otherwise compromise the structural integrity of the nucleosome during ATP-dependent repositioning.

The intrinsic ability of the nucleosome to transiently buffer additional bp of DNA may represent a common property harnessed by other chromatin-interacting machinery, and thus would more broadly influence chromatin reorganization during processes such as transcription, DNA replication, and repair.

## Methods

**Preparation of fluorophore-labelled mononucleosomes.** All nucleosome constructs used the Widom 601 nucleosome positioning sequence[53] and had asymmetric lengths of flanking DNA. Double-stranded DNA was prepared by annealing (in equimolar ratios) and ligating (T4 DNA ligase, New England Biolabs) a set of complementary, overlapping oligonucleotides. HPLC-purified oligonucleotides with required modifications were obtained from Integrated DNA Technologies (IDT). The internal Cy3 dye (iCy3) was backbone incorporated, replacing an entire nucleotide and thus resulting in an unpaired base on the opposite strand. The Alexa750 dye was attached to position 5 of a dT base via a 6-carbon linker. All DNA constructs were purified by PAGE. The sequences of the DNA constructs and oligonucleotides they were assembled from are provided in Supplementary Tables 1 and 2, respectively.

Nucleosomes were made using *Xenopus laevis* histones[74,75]. All histones were expressed in *E. coli* BL21 (DE3) pLysS cells, grown in 4 L batches of 2x TY media containing 100 μg/mL ampicillin and 34 μg/mL chloramphenicol. Cultures were grown at 37 °C to OD$_{600}$ ~0.4 before inducing overexpression with addition of 0.3 mM IPTG. Expression continued for 3 h before cells were recovered by centrifugation at room temperature. Cell pellets were resuspended in ~30 mL wash buffer (50 mM Tris-HCl pH 7.5, 100 mM NaCl, 1 mM EDTA, 1 mM benzamidine (added fresh)) before storing at −80 °C. For each 4 L growth, the cell pellet was thawed, resuspended in ~80 mL of wash buffer, and sonicated on ice. The lysate was centrifuged at 4 °C for 20 min at 23,000 × g. The supernatant was decanted and the pellet containing inclusion bodies of the histone protein was resuspended in wash buffer + detergent (1% v/v Triton X-100). Centrifugation and washing with wash buffer + detergent was repeated once more followed by two more washes with wash buffer. After the final wash, the supernatant was discarded and the white inclusion body pellet was spread in a thin layer over the inside of a 50 mL Falcon tube and stored at −20 °C. Chromatography buffers containing urea were prepared the morning of histone purification to reduce the formation of isocyanate, which can modify protein residues. To remove isocyanate, urea was first dissolved in ultra-pure water and treated with AG 501-X8(D) resin (Bio-Rad) before adding additional buffer components. All chromatography buffers were 0.2 μm filtered and degassed for at least 30 min. The inclusion body pellet was treated with 1 mL DMSO and agitated for 30 min at room temperature. Unfolding buffer (7 M guanidine-HCl, 20 mM Tris-HCl pH 7.5, 1 mM EDTA, 10 mM DTT) was added to the pellet up to a 40 mL total volume and agitated at room temperature for 1 h. The soluble, unfolded protein was separated from the insoluble fraction by centrifugation at 23,000 × g for 20 min at 18 °C and exchanged into urea buffer A (10 mM Tris-HCl pH 7.5, 7 M urea, 1 mM EDTA, 100 mM NaCl, 5 mM β-mercaptoethanol) by passages of <25 mL over a desalting column (GE HiPrep 26/10, 17-5087-01). The protein solution was then applied to two tandem ion exchange columns: a HiPrep 16/10 Q FF (GE Healthcare 17-5190-01) anion exchange column followed by a HiPrep 16/10 SP FF (GE Healthcare 17-5192-01) cation exchange column. After loading, the Q FF column was detached and the SP FF column was washed further until UV absorbance reached baseline. Histone proteins were then eluted from the SP FF column with a gradient of buffer B (10 mM Tris-HCl pH 7.5, 7 M urea, 1 mM EDTA, 1 M NaCl, 5 mM β-mercaptoethanol) from 0–50% over 30 column volumes. Fractions containing histones were analyzed by SDS PAGE (18% acrylamide), pooled, and then dialyzed extensively into water with 5 mM β-mercaptoethanol using at least three 4 L dialyzations. For each histone, the final pool was divided into ~2 mg aliquots, lyophilized, and then stored at −20 °C until needed.

To produce fluorescently labelled histones, the single-cysteine variants H2A (T120C) and H2B(K120C) were separately reacted with Cy5-maleimide[76]. Each 2 mg aliquot of lyophilized histone was dissolved in 1.0 mL of labeling buffer (20 mM Tris-HCl pH 7.0, 6 M guanidine-HCl, 5 mM EDTA) and cysteines were reduced by adding 4 μL of 500 mM tris(2-carboxyethyl)phosphine (TCEP). After unfolding for 2 h at room temperature, Cy5-maleimide was added to a concentration of 3 mM and incubated in the dark for 3 h at room temperature. The labeling reaction was quenched with 80 mM β-mercaptoethanol. Excess dye was removed by washing the histones with labeling buffer ≥5 times in a 4 mL Amicon Ultra 10,000 MWCO concentrator before bringing the volume to 1.5 mL. The labeling efficiency of the Cy5-labelled histones was approximately 70–85%.

Unlabelled histones were unfolded by adding 1.5 mL unfolding buffer to each 2 mg aliquot and incubated at room temperature for 1–3 h. Each fluorescently labelled histone was combined in equimolar amounts with either unlabelled H2A or H2B to generate H2A/H2B dimers. Wild-type *Xenopus* histone H4 and H3 (C110A) were combined in equimolar amounts to form (H3/H4)$_2$ tetramers. Histone dimers and tetramers were refolded separately in 3500 MWCO dialysis tubing using four dialysis steps of at least 4 h in 500 mL of refolding buffer (10 mM Tris-HCl pH 7.5, 2 M NaCl, 1 mM EDTA, 5 mM β-mercaptoethanol). Histone dimers and tetramers were each purified by size exclusion chromatography over a

HiLoad 16/600 Superdex 200 pg column (GE Healthcare, 28989335) pre-equilibrated in refolding buffer. After analysis on 18% SDS PAGE gels, fractions with equal amounts of histones were pooled, brought to 20% glycerol, frozen in liquid N$_2$, and stored at −80 °C.

Nucleosomes and hexasomes were reconstituted by deposition of histone dimers and tetramers onto labelled DNA using the standard salt gradient dialysis technique[75]. In this technique, the histone components are added to DNA in the presence of 2 M KCl, and then dialyzed against 400 mL of high-salt reconstitution buffer (10 mM Tris-HCl pH 7.5, 2 M KCl, 1 mM EDTA, 1 mM DTT (added fresh)) at 4 °C. To generate a dialysis gradient, a pump is set up to remove the high-salt buffer while simultaneously adding in 2 L of low salt buffer (10 mM Tris-HCl pH 7.5, 250 mM KCl, 1 mM EDTA, 1 mM DTT (added fresh)) at ~1 mL/min. To enrich for hexasomes over nucleosomes, H2A/H2B dimers were added to H3/H4 tetramers at approximately a 1.2:1 ratio, and hexasome and nucleosome species were purified away from each other using a BioRad MiniPrep or Prep Cell apparatus[54,75]. Native acrylamide columns were poured containing 7% polyacrylamide (60:1 acrylamide:N,N-methylene-bis-acrylamide) and 0.75 X Tris-Borate-EDTA buffer (TBE). MiniPrep Cells or 28 mm diameter Prep Cells were poured to heights of 7 cm and 6 cm, respectively. Nucleosome reconstitutions were concentrated to ~50 μL and ~250 μL for the MiniPrep Cell and Prep Cell, respectively, and brought to 5% glycerol to facilitate loading. purifications were conducted at 4 °C using degassed 0.5 X TBE as running buffer and products were eluted in 10 mM Tris-HCl pH 7.5, 1 mM EDTA, 1 mM DTT. MiniPrep Cells and Prep Cells were run at 1 and 10 watts, respectively, and nucleosome species typically eluted in ~5–6 h depending on construct and purification apparatus. Fractions were analyzed by native PAGE, pooled and frozen in 20% glycerol. The purified hexasome pools were used for single-molecule experiments, with addition of labelled or unlabelled H2A/H2B dimers resulting in singly labelled nucleosomes[54].

**Expression and purification of Chd1 and SNF2h.** Chd1 (*S. cerevisiae*, residues 118–1274) and SNF2h (*H. sapiens*, full length) were expressed in *E. coli* and purified using similar procedures[58,77,78]. pDEST17 plasmid containing Chd1 was transformed into BL21(DE3) Trigger RIL *E. coli* cells and cultures were grown in TB media at 37 °C. pBH4 plasmid containing SNF2h was transformed into BL21 (DE3) R* *E. coli* cells and cultures were grown in 2X LB (1X NaCl) at 37 °C. After induction at OD$_{600}$ of ~0.6 with 0.3 mM IPTG and expression at 18 °C over 16 hr, cells were harvested by centrifugation and then lysed on ice by sonication and addition of lysozyme, and after clarification by centrifugation, protein was purified by passage over a NiNTA resin (2–3 tandem 5 mL HisTrap columns, GE Healthcare, 17-5248-01) followed by ion exchange chromatography (5 mL HiTrap SP FF, GE Healthcare, 17-5054-01). The 6xHis tag was removed by protease digestion (prescission protease for Chd1 and TEV for SNF2h) overnight at 4 °C, and protein was further purified by repassage over the NiNTA resin and subjected to S200 size exclusion chromatography (HiLoad 16/600 Superdex 200 pg, GE Healthcare, 28989335). The purified protein was concentrated and stored in small aliquots at −80 °C. Thawed aliquots of remodeler protein were always kept on ice and used within 12 h for each experiment.

**Single-molecule FRET.** Biotinylated and fluorophore-labelled mononucleosomes were surface-immobilized on PEG (poly[ethylene glycol])-coated quartz microscope slides through biotin-streptavidin linkage[38], which did not interfere with the remodelling activity[41]. Cy3 and Cy5 fluorophores were excited with 532 nm Nd: YAG and 638 nm diode lasers, respectively, and fluorescence emissions from Cy3, Cy5, and Alexa750 were detected using a custom-built prism-based TIRF microscope, filtered with ZET532NF (Chroma) and NF03-642E (Semrock) notch filters, spectrally separated by 635 nm (T635lpxr) and 760 nm (T760lpxr) dichroic mirrors (Chroma), and imaged onto the three thirds of an Andor iXon Ultra 888 EMCCD camera. For three-colour FRET imaging, an alternating laser excitation was achieved by switching the 532 nm and the 638 nm lasers using mechanical shutters. Data acquisition was controlled using MicroManager[79]. Fluorescence emission time traces were corrected to account for the direct excitation of Cy5 by the 532 nm laser and of Alexa750 by the 638 nm laser (direct excitation of Alexa750 by the 532 nm laser was negligible), as well as for the bleedthrough from the Cy3 into the Cy5 channel and that from Cy5 into the Alexa750 channel. Cy3 and Alexa750 signals were scaled to correct for the differences in quantum yields and detection efficiencies between the three dyes (see Supplementary Note 1 and Supplementary Fig. 9 for further details). Cy3-Alexa750 FRET is impossible to distinguish from Cy3-Cy5 FRET when Cy5-Alexa750 FRET is close to 1. Therefore, Cy3-Cy5 and Cy3-Alexa750 FRET values were reported as an entry-side (for nucleosomes in Supplementary Fig. 1) or exit-side (for all other nucleosomes) FRET value that corresponds to all FRET originating from Cy3, calculated as

$$1 - \frac{I_{Cy3}^{532}}{I_{Cy3}^{532} + I_{Cy5}^{532} + I_{A750}^{532}} \qquad (1)$$

where $I_{Cy3}^{532}$, $I_{Cy5}^{532}$, and $I_{A750}^{532}$ are fluorescence intensities upon excitation with the 532 nm laser of Cy3, Cy5, and Alexa750, respectively. Exit-side (for nucleosomes in Supplementary Fig. 1) or entry-side (for all other nucleosomes) FRET was

calculated as

$$\frac{I^{638}_{A750}}{I^{638}_{Cy5} + I^{638}_{A750}} \tag{2}$$

where $I^{638}_{Cy5}$ and $I^{638}_{A750}$ are the fluorescence intensities upon excitation with the 638 nm laser of Cy5 and Alexa750, respectively. For nucleosomes with internally positioned fluorophores (labeling schemes in Figs. 1a and 2d), the initial Cy3-Alexa750 distance of >8 nm, the relatively small Cy3-Alexa750 Förster radius of ~3.8 nm[80], and the geometric constraints of the nucleosome structure together resulted in the absence of any detectable Cy3-Alexa750 FRET. Accordingly, typical time traces of nucleosomes with bleached Cy5 dye did not exhibit any detectable Cy3-Alexa750 FRET (Supplementary Fig. 3a). Data were analysed using the Fiji distribution of ImageJ[81,82], IDL, and Matlab. The lag time in the individual traces was determined by visual inspection, and exponential fitting to the lag-time distribution was used to determine the mean lag times. Imaging experiments were carried out in imaging buffer containing 40 mM Tris pH 7.5, 12 mM HEPES pH 7.9, 60 mM KCl, 0.32 mM EDTA, 3 mM MgCl₂, 100 µg/mL acetylated BSA (Promega), 10% (v/v) glycerol, 10% (w/v) glucose, 2 mM Trolox to reduce photoblinking of the dyes[83], as well as an enzymatic oxygen scavenging system (composed of 800 µg/mL glucose oxidase and 50 µg/mL catalase). Using a syringe pump (Harvard Apparatus), remodelling was initiated by infusing the sample chamber with imaging buffer supplemented with remodeller and ATP or ATP-γ-S as well as additional MgCl₂ equimolar to the total amount of added nucleotide. Data for FRET distribution histograms were collected after at least 10 min of incubation in the presence of the remodeller alone or with the specified nucleotide, sufficiently long for remodelling (in the case of ATP) or binding (in the absence of nucleotide or presence of ATP-γ-S) to reach equilibrium.

**Site-specific cross-linking.** DNA movement on gapped nucleosomes was monitored during remodelling reactions using site-specific histone-to-DNA cross-linking assays as previously described with some modifications[62,63]. Nucleosomes with 5' fluorescent DNA labels (Cy5-4-601-79-FAM) were generated containing single cysteines at both the dyad (H3 M120C) and entry/exit regions (H2B S53C). Nucleosomes were buffer exchanged into 20 mM Tris-HCl pH 7.5, 5% (v/v) glycerol to remove DTT, and cysteines were labelled with the photoactivatable cross-linker, 4-azidophenacyl bromide (APB, ~220 µM C_f) at room temperature in the dark for 2–3 h. Labelling reactions were subsequently quenched with DTT. Nucleosome (150 nM) and Chd1 (200 nM) were pre-incubated for 10–15 min at room temperature in slide buffer (20 mM Tris-HCl pH 7.5, 50 mM KCl, 5 mM MgCl₂, 5% (w/v) sucrose, 0.1 mg/mL BSA, 1 mM DTT). Sliding reactions (50 uL) were initiated by addition of 1 mM ATP (1 mM C_f). A reaction time-course was collected (0", 7", 15", 30", 45", 60", 90", 2', 4', 8', 16', 32'), wherein at the appropriate time, each reaction was irradiated at 302 nm under a UV transilluminator (VWR) for 10 s to induce cross-linking of APB to the DNA. Each reaction was diluted with addition of 100 µL slide buffer, and DNA was dissociated from histones through addition of post-irradiation buffer (20 mM Tris-HCl pH 8.0, 0.2% SDS, 50 mM NaCl) and heating at 70 °C for 20 min. DNA was extracted from reactions by addition of 300 µL 5:1 phenol:chloroform. Cross-linked DNA, located at the organic/aqueous interphase, was enriched relative to free DNA by washing the aqueous phase 3–4 times with 1 M Tris-HCl pH 8.0, 1% SDS. DNA was ethanol precipitated to remove phenol and resuspended in 100 µL of resuspension buffer (20 mM ammonium acetate, 2% SDS, 0.1 mM EDTA pH 8.0). Alkaline cleavage of the DNA at the cross-linked site was stimulated by addition of 5 µL of 2 M NaOH and heating at 95 °C for 40 min. DNA was ethanol precipitated and resuspended in formamide gel loading dye (89 mM Tris-borate pH 8.0, 5 mM EDTA, 95% (v/v) deionized formamide, 0.2% (w/v) Orange G dye). The DNA fragments were separated alongside a Sanger sequencing ladder on an 8% polyacrylamide (19:1 acrylamide:N,N-methylene-bis-acrylamide), 8 M urea sequencing gel run at 65 W for 1.5 h. The gels were scanned on a Typhoon 9410 variable mode imager (GE Healthcare) and gel band densities were plotted using ImageJ software[81]. Uncropped gels are provided in the accompanying Source Data file.

**Native gel electrophoretic mobility assay.** Nucleosome sliding reactions were monitored by the differential migration of translational states of the nucleosome using native gels as described previously[84]. Reactions were designed to match the conditions in the site-specific cross-linking assay and thus contained 150 nM nucleosome and 200 nM Chd1 in slide buffer (20 mM HEPES-KOH pH 7.6, 50 mM KCl, 5 mM MgCl₂, 5% (w/v) sucrose, 0.1 mg/mL BSA, 1 mM DTT). Nucleosomes and Chd1 were pre-incubated at room temperature for 10–15 min before starting the sliding reaction with 1 mM ATP. The reaction progress was monitored by quenching portions of the reaction with 25 mM EDTA and 1 µg/µL salmon sperm DNA at timepoints (0", 15", 30", 1', 2', 4', 8', 16', 32', 64'). Samples were loaded on 6% polyacrylamide (60:1 acrylamide: N,N-methylene-bis-acrylamide) 0.75X TBE gels while running at 100 V with 0.25X TBE at 4 °C. Gels were run for 1.75 h at 130 V and scanned on a Typhoon 5 variable mode imager (GE Healthcare).

**Reporting summary.** Further information on experimental design is available in the Nature Research Reporting Summary linked to this article.

## Data availability
Data supporting the findings of this manuscript are available from the corresponding authors upon reasonable request. A reporting summary for this Article is available as a Supplementary Information file. The source data underlying Figs. 1–3 and Supplementary Figs. 2–8 are provided as a Source Data file.

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

## Acknowledgements

S.D. acknowledges support from a European Research Council (ERC) Starting Grant (ChromatinRemodelling), the Swedish Research Council (VR 2015-04568), the Knut and Alice Wallenberg Foundation (KAW/WAF 2014.0183), and the Science for Life

Laboratory (SciLifeLab). G.D.B. is supported by the National Institutes of Health (R01-GM084192 and R01-GM113240). X.Z. is a Howard Hughes Medical Institute Investigator. S.D. is an EMBO Young Investigator.

## Author contributions

S.D. and X.Z. conceived of the original 3-colour FRET remodelling experiment to test coordination of DNA movements on the entry-side and exit-side of nucleosomes during remodelling; A.S., R.F.L., G.D.B., and S.D. designed the 3-colour nucleosomes; A.S. and R.F.L. produced all reagents; A.S. carried out all single-molecule experiments; A.S. and S.D. analyzed single-molecule experiments; R.F.L. carried out nucleosome cross-linking experiments; S.D. and G.D.B. oversaw the project; A.S., R.F.L., G.D.B., and S.D. wrote the manuscript, with input from X.Z.

## Additional information

**Competing interests:** The authors declare no competing interests.

