## [Peer Review File · Nature Communications]

Reviewers' Comments:

Reviewer #1:

Remarks to the Author:

The manuscript "Direct observation of coordinated DNA movements on the nucleosome during chromatin remodelling" by Sabantsev and co-workers reports on the discontinuous movement of nucleosome bound DNA induced by the ATP driven remodeler Chd1. Using three-colour smFRET and cross linking experiments, the authors demonstrate that the nucleosome is able to "buffer" DNA, as exemplified in different timescales of the DNA movement and repositioning of entry-side and exit-side DNA. The proposed model suggests that interactions between remodellers lead to coordinated remodelling whilst maintaining the structural integrity of the nucleosome. The manuscript is technically well written, but somewhat difficult to follow especially for readers not familiar with the research field or some of the elaborate techniques used.

Some main points I would like to see discussed

- 1) In the second part of the manuscript, the authors use gapped DNA substrates to limit DNA translocation, but I keep wondering to which extent this huge modification (gapped ends of DNA can fray by a few base pairs) still allows to draw valid conclusions for the native case of fully duplexed DNA?
- 2) In fact, the authors see "structural perturbations" with their smFRET sensor leading to the outcome that smFRET is not suitable for further characterisation of the DNA movements. Could the authors elaborate what they mean here with structural perturbations?
- 3) The authors then continue with the cross-linking experiments, but it is not clear to me why these experiments are considered a valid alternative to the smFRET experiments.

Minor points and other suggestions

- 1) Fig 1. The DNA scheme 1 is introduced in an entire paragraph only to get replaced by the better suited scheme 2 shortly after. Maybe shift entirely to SI. Instead, I would like to see the 2D representation of scheme 2 added to Fig 1. as it helped me a lot to see what is going on (hexasome and dimers are not visible in the 3D representation, although Chd1 should be added to the 2D presentation).
- 2) Fig 1. Why not mentioning the number of molecules in c) and d) directly instead of first saying "many" and then >400 in the figure caption?
- 3) Fig 1. What was the time between the measurements with and without ATP or were those independent, equilibrated experiments?
- 4) Fig. 2. How was the lag time determined in the individual traces? Visual inspection? How was the mean lag time determined? Exponential fit over a distribution of lag times?
- 5) See above, the manuscript suddenly switches from FRET measurements to cross-linking experiments. As someone with little experience in those, 1-2 sentences on what the idea behind these experiments is and how they are then executed would be helpful. E.g. what (Cys) is linked to what (DNA bases) and why is this relevant (results in ssDNAs of different lengths that are then separated on a gel allowing to judge which base was closest to the cysteine residue, correct?)
- 6) Fig.2c time course values of the experiment could be directly added to the figure.
- 7) Fig.3c. Maybe the authors can indicate the several subunit of the complex.

8) Three-colour FRET is extremely challenging. What was the labelling efficiency of the histones? The authors mentioned that the intensity values of the time traces were scaled to correct for quantum, detection efficiencies etc, but no details on these values or the scaling factors were given. Could the authors add some uncorrected time traces in the supplement allowing to judge the raw data?

Reviewer #2:

Remarks to the Author:

With perhaps only one exception, every characterized ATP-dependent chromatin remodeling enzyme can use the energy of ATP hydrolysis to re-position or slide nucleosomes in cis. In the past few years, many studies have exploited histone-DNA crosslinking, ensemble, and single molecule FRET methods to follow the movement of nucleosomes by remodelers. In all cases, such studies have followed how the remodeler "pulls-in" or "pushes-out" DNA from the nucleosome during the translocation reaction. Such studies have yielded major insights into the DNA translocation steps that drive nucleosome positioning. For instance, we know that remodelers translocate 1bp of DNA for each ATP hydrolyzed, and remodelers such as SWI/SNF or RSC can drive bursts of translocation that can generate loops on the nucleosome surface. As a single subunit enzyme, the yeast Chd1 has developed into a powerful model for studies of remodeling enzymes. Work from the Bowman group has been instrumental in showing how this enzyme re-positions nucleosomes and how its activity is regulated by its associated chromodomains and nucleosomal linker DNA.

One key question that has not been directly addressed by previous studies is how does the translocation of DNA at a fixed position on one side of the nucleosome get propagated to the opposite side, leading to sliding? Previous work with ISWI family members monitored the independent rates for movement of DNA "into" and "out" of the nucleosome, and they concluded that DNA actually exited the nucleosome prior to DNA being "pulled-in". However, these two events were not measured simultaneously, and thus the conclusion was indirect. In this manuscript by Bowman and colleagues, the authors have performed three-color, single molecule FRET experiments that simultaneously monitor movements of DNA at both the entry and exit positions. To my knowledge this is the first example where such an experiment has been performed for remodeling enzymes. The results are unequivocal -- Chd1 clearly translocates 1-3 bp of DNA from the entry side of the nucleosome, but this DNA does not immediately protrude from the other, exit side until 1 or more additional ATPs have been hydrolyzed (presumably until 1 or more bps have been translocated). Thus, there is a clear time delay between pulling DNA in and its exit from the nucleosome. The authors reinforce their smFRET data with histone-DNA crosslinking studies with gapped substrates. The authors propose a very compelling model in which the 1-3 bp of DNA is "buffered" within the nucleosome, following the first rounds of DNA translocation. Subsequent rounds of translocation drives this DNA from the exit side of the nucleosome. The authors propose that the ability to buffer, or hold, extra DNA within the nucleosome is likely to regulate the ability of a second copy of Chd1 to engage the nucleosome from the opposite side.

In general, the data presented are extremely high quality and the interpretation of data balanced. Every possible control for the FRET studies have been performed, so I have no technical concerns. My only criticism relates to the authors' estimate that 1-3 bp of DNA is buffered by the nucleosome. This range of values is based on the fact that a substrate with a 2nt gap located 5 bp from SHL2 (m-5) does not block movement of DNA from the exit side of the nucleosome. Although their conclusion is correct, they could in principle determine exactly how much DNA can be buffered by the nucleosome during this reaction. This may seem like a trivial point, but this buffering phenomenon is perhaps the main point of the paper, so defining the # of bps is key. I realize it is technically quite a bit of work, but repeating the crosslinking studies with the m=4,3,2,1 substrates (or a subset) would solidify this main point.

Related minor point:

The authors show with smFRET analyses that a $m=0$ substrate (2nt gap at SHL2) leads to aberrant changes in FRET due solely to Chd1 binding. This seems very odd and the authors don't really follow this up. I was surprised that they did not try the $m=8$ substrate by smFRET or try the $m=0$ substrate with the crosslinking assay. It may be that an $m=3$ or $m=4$ substrate would yield interpretable results by smFRET. If so, this would boost the conclusions. In general it would be good to have overlap between these two different assays.

Craig Peterson

Reviewer #3:

Remarks to the Author:

The paper by Sabantsev et al. describes experiments studying the movements of DNA on nucleosomes under the action of the remodeler protein Chd1. They use an advanced 3-color single molecule FRET assay to report separately on the movement of both ends of the DNA on either side of the nucleosome relative to one of the histone proteins. The key finding is that the remodeling protein Chd1 causes the DNA on the entry side of the nucleosome to shift before the exit side DNA shifts. This time delay is dependent on the ATP concentration, suggesting there is a step that requires ATP binding from solution. This temporal delay in DNA shifting on entry and exit sides requires the nucleosome to transiently accommodate (or buffer in the authors words) additional DNA. The authors vary the label position and perform cross-linking studies with gapped DNA to conclude that the nucleosome must accommodate 1-3 extra basepairs during the delay between entry side and exit side motion. Finally, they suggest that this buffering provides a mechanism for communication between remodeling proteins on opposite sides of the nucleosome.

This is a clever application of multiple FRET reporters on one sample. The experiments appear expertly done. The paper is clear and well written. It is an interesting result and the experimental findings strongly support the conclusions. I do not have the experience or expertise in the biochemistry of chromatin remodeling enzymes to comment on the details of the literature background and impact in that field. For that aspect, I ask the editors to rely on the other reviewers. Based on the expertly performed, creative experiments along with the generally interesting result of characterizing a mechanistic interaction of a remodeling protein moving a nucleosome by first moving DNA on one side and then later on the other side, forcing a transient absorption of extra DNA, I support publication. I only have a few small comments for the authors to consider.

1. The authors determine that gapped DNA causes enough structural problems to prevent FRET studies (figure supplemental fig 6), but the authors trust the conclusions of the crosslinking work (figure supplemental figure 7 and fig 3). The authors should add some discussion about why they believe the structural problems caused by gapped DNA identified in the attempted FRET study do not impact the relevance of the conclusions from the crosslinking study.

2. On page 7, at the beginning of the section "The onset of exit-side movement involves an additional ATP-binding event", the authors state that "At subsaturating ATP concentrations...". It would be helpful for non-experts to evaluate the relevance of their data if the authors cite and mention biochemical literature on the ATP affinity of the ATPase in Chd1. What are subsaturating or saturating concentrations for this enzyme?

3. On page 15, in single-molecule FRET section of methods, it is stated that Cy3 and Alexa750 signals are scaled to correct for differences in QY and detection efficiency. Some readers would be interested to know those scaling factors and they could be added to the methods.

We appreciate the time and energy that Reviewers have taken in evaluating our manuscript, and were happy to see positive remarks from all three Reviewers. In addition to text changes suggested by the Reviewers, we have added the following additional experiments to strengthen the manuscript:

1. To extend the three-color smFRET analysis of the $m=0$ bp gapped nucleosome, we generated and tested $m=3, 5,$ and 8 bp gapped nucleosomes (Supplementary Fig. 5). These gapped nucleosomes showed the same trends as $m=0$ bp nucleosomes, producing Chd1-dependent FRET changes without nucleotide (apo) and/or with ATPyS.
2. We generated and tested cross-linking of an $m=3$ bp gapped nucleosome, which failed to be repositioned by Chd1 (Supplementary Fig. 6).
3. To characterize whether Chd1 interactions with the $m=3$ bp gapped nucleosome might be different from the others, we cross-linked the Chd1(N650C) variant to the $m=3, 5, 8$ bp gapped and also an ungapped nucleosome (Supplementary Fig. 8). Interestingly, Chd1 cross-linked more strongly to the $m=3$ bp gapped nucleosome than the others, suggesting that interactions between DNA and the ATPase motor may have been perturbed.
4. To determine whether the delay in exit-side DNA movement might be a general phenomenon, we performed three-color smFRET experiments using the ISWI-type remodeler SNF2h (Fig. 2e). Strikingly, the same order of DNA movement - entry first, exit second - was observed for SNF2h nucleosome sliding reactions.

Below is a point-by-point response to each Reviewer's comments.

Reviewer #1:

1) In the second part of the manuscript, the authors use gapped DNA substrates to limit DNA translocation, but I keep wondering to which extent this huge modification (gapped ends of DNA can fray by a few base pairs) still allows to draw valid conclusions for the native case of fully duplexed DNA?

Multiple groups have shown that nucleosomes assemble properly in the presence of single-stranded DNA gaps, and that chromatin remodelers can shift gapped nucleosomes, albeit to a limited extent. We agree with the Reviewer's concerns that a single-stranded gap is a significant modification to the DNA, and exactly how a gap stops the ATPase (whether through fraying or the absence of DNA backbone) is not clear. However, our purpose in this work was to see to what point limited entry side DNA movement may be transferred (or not) all the way to the exit side. In our design, the gap should not interfere with propagation or buffering of ~ 90 bp of DNA (entry SHL2 all the way around to exit side) around the histone octamer due to its location: the gap is on the entry side of the ATPase motor binding site, and therefore whatever DNA the ATPase shifts towards the dyad should be effectively insulated from perturbations around the gap.

2) *In fact, the authors see “structural perturbations” with their smFRET sensor leading to the outcome that smFRET is not suitable for further characterisation of the DNA movements. Could the authors elaborate what they mean here with structural perturbations?*

As we describe in the revised manuscript, we were concerned that the combination of an unpaired DNA nucleotide opposite the backbone-incorporated dye along with a single-stranded gap may have been causing some structural perturbations. ATP-independent changes were observed to varying extents for all the gapped nucleosomes tested, namely for the $m= 0, 3, 5$ and 8 bp gap locations (see Supplementary Fig. 5). We did not observe such ATP-independent changes in the dyes on ungapped nucleosomes, therefore the DNA gaps contributed to this effect.

The following text has been included in the revised manuscript to make this point clear to the reader:

“While internal dyes allow for ATP-dependent nucleosome sliding, we note that such backbone-incorporated labels result in an unpaired nucleotide opposite the dye. The presence of a 2 nt DNA gap at the entry SHL2, in combination with the unpaired nucleotide from the label, may give rise to structural perturbations of DNA that produce FRET changes independently of ATP hydrolysis. Further, FRET is sensitive to transient, non-canonical placements of the DNA on the nucleosome, which could obscure detection of ATP-dependent DNA translocation.”

3) *The authors than continue with the cross-linking experiments, but it is not clear to me why these experiments are considered a valid alternative to the smFRET experiments.*

As we describe in the quoted manuscript text above, investigating this system using cross-linking rather than smFRET provides two main advantages. First, internal dyes are not required for cross-linking, and therefore any potential unwanted effects from unpaired DNA bases combined with the single-stranded gaps are avoided. Second, cross-linking has particular advantages over smFRET for confidently determining DNA translocation (compared with other possible structural changes). In our experience, cross-linking only reports on DNA lying in the canonical DNA path around the histone octamer. Therefore, transient excursions away from the canonical nucleosome structure would reduce the overall amplitude of observed cross-links, but not alter the DNA translocation readout. FRET on the other hand, is much more sensitive to structural perturbations, and non-canonical intermediate structures are expected to give rise to a complex FRET signature, obfuscating the readout of how DNA changes position.

Minor points and other suggestions

1) *Fig 1. The DNA scheme 1 is introduced in an entire paragraph only to get replaced by the better suited scheme 2 shortly after. Maybe shift entirely to SI. Instead, I would like to see the 2D representation of scheme 2 added to Fig 1. as it helped me a lot to see what is going on (hexasome and dimers are not visible in the 3D representation, although Chd1 should be added to the 2D presentation).*

We agree with the Reviewer's suggestion of changing the figure, and have moved the original "scheme 1" to the supplement and replaced it with the 2D representation (Fig. 1a) of how nucleosomes were assembled: "hexasome+dimer=nucleosome." As the 2D representation of Figure 1a is rather busy, we refrained from showing Chd1 there, although we did add it to the analogous diagram in Figure 3c. While we altered the text describing the two schemes for three-color labeling, we felt it was useful to still briefly describe our first attempt at using a more traditionally labeled nucleosome, since that construct is related to what others in the field have used for two-color FRET.

2) Fig 1. Why not mentioning the number of molecules in c) and d) directly instead of first saying "many" and then >400 in the figure caption?

We apologize for this inconsistency; the revised manuscript now provides the number of observations.

3) Fig 1. What was the time between the measurements with and without ATP or were those independent, equilibrated experiments?

The measurements were taken on the same slides before and at least 10 minutes after the addition of Chd1 with ATP. This amount of time was sufficient for the remodeling process to reach equilibrium. We have now revised the methods section to more clearly state this.

4) Fig. 2. How was the lag time determined in the individual traces? Visual inspection? How was the mean lag time determined? Exponential fit over a distribution of lag times?

The lag time in the individual traces was determined by visual inspection, and exponential fitting to the lag time distribution was used to determine the mean lag times. The methods section of the manuscript was updated to clarify this.

In light of the reviewer's comment, we have also carried out an automated detection of lag times using a representative subset of the data (> 200 traces showing nucleosome remodeling by Chd1 at 100 μ M ATP), which yielded essentially the same results (see Figure below). Briefly, the onset of entry-side and exit-side movement was automatically determined as the first time point where the corresponding fluorescence intensities changed from the initial values in the direction expected for remodeling above a threshold based on the standard deviation of the initial part of the trace (before the addition of the remodeler).

5) See above, the manuscript suddenly switches from FRET measurements to cross-linking experiments. As someone with little experiments in those, 1-2 sentences on what the idea behind these experiments is and how they are then executed would be helpful. E.g. what (Cys) is linked to what (DNA bases) and why is this relevant (results in ssDNAs of different lengths that are then separated on a gel allowing to judge which base was closest to the cysteine residue, correct?)

We apologize for not explaining this technique more clearly before. To address this point, we now briefly describe how the cross-linking works this way in the manuscript:

“With this technique, single cysteines are labeled with a photo-reactive cross-linker, which can produce nicks in the DNA backbone at sites of cross-linking (Kassabov et al., 2003; Patel et al., 2013; Winger et al., 2018). Shifts in cross-linking positions, monitored on urea denaturing gels, are interpreted as relative displacements in DNA past the cross-linking site on the histone core. To detect DNA movement at several sites simultaneously, we produced nucleosomes with two distinct cysteines: H2B(S53C), which cross-links on the outer gyre of DNA and therefore reports on both the entry and exit side, and H3(M120C), which forms dual cross-links around the nucleosome dyad (Fig. 3c).”

6) Fig.2c time course values of the experiment could be directly added to the figure.

We are not sure if this is the same panel that the Reviewer was referring to, but we realized that time course values were omitted for Fig. 3b. This has now been corrected.

7) Fig.3c. Maybe the authors can indicate the several subunit of the complex.

The Chd1 remodeler has just one subunit, but we only illustrate the ATPase motor domains in Fig. 3a and 3c for clarity, as it is unclear how other domains of Chd1 participate in nucleosome sliding. To help clarify where the cross-links are coming from, we have added the H2B(S53C) and H3(M120C) positions on the molecular representation in Fig. 3a, which is meant to complement the 2D representation in Fig. 3c.

8) Three-colour FRET is extremely challenging.

What was the labelling efficiency of the histones?

The labeling efficiency of the Cy5-labeled histones were approximately 70-85% (as determined by measuring the absorptions at 280 and 650 nm). As we describe in the text, we first made hexasomes with unlabeled H2B; then, the labeled H2B (folded together with H2A) was added to transform the hexasomes to nucleosomes. With this stepwise procedure, there should be at most one labeled H2B for each nucleosome, and single-molecule observations agreed with the estimated 70-85% labeling efficiency. The DNA labels were incorporated by IDT during oligo synthesis, and we therefore considered H2B-Cy5 to be the most limiting dye. Nucleosomes devoid of any Cy5 label were easily identified by direct excitation with the red laser.

The authors mentioned that the intensity values of the time traces were scaled to correct for quantum, detection efficiencies etc, but no details on these values or the scaling factors were given. Could the authors add some uncorrected time traces in the supplement allowing to judge the raw data?

We apologize for this oversight and have now included in the revised manuscript details on the correction for differences in quantum yields and detection efficiencies as well as bleed-through and direct excitation (Supplementary Note 1). To further illustrate these corrections and following the reviewer's suggestion, we have included a new supplementary figure showing a direct comparison of raw and corrected time traces (Supplementary Fig. 9).

Reviewer #2:

In general, the data presented are extremely high quality and the interpretation of data balanced. Every possible control for the FRET studies have been performed, so I have no technical concerns. My only criticism relates to the authors' estimate that 1-3 bp of DNA is buffered by the nucleosome. This range of values is based on the fact that a substrate with a 2nt gap located 5 bp from SHL2 ($m=5$) does not block movement of DNA from the exit side of the nucleosome. Although their conclusion is correct, they could in principle determine exactly how much DNA can be buffered by the nucleosome during this reaction. This may seem like a trivial point, but this buffering phenomenon is perhaps the main point of the paper, so defining the # of bps is key. I realize it is technically quite a bit of work, but repeating the crosslinking studies with the $m=4,3,2,1$ substrates (or a subset) would solidify this main point.

In light of the Reviewer's comment, we have now additionally examined nucleosome sliding with $m=3$ bp gapped nucleosomes and included these new data in the revised manuscript. In native gel sliding assays, we did not observe any apparent Chd1-induced repositioning of the $m=3$ bp gapped nucleosomes (Fig. 3b of the revised manuscript). Similarly, cross-linking experiments with nucleosomes featuring a gap at the $m=3$ bp location showed no apparent shifts in the observed cross-links after incubation with Chd1 and ATP (Supplementary Fig. 6 of the revised manuscript).

Given the 3-4 bp translocation observed for the $m=5$ bp gapped nucleosome, it is unclear why the $m=3$ bp nucleosomes, with the gap relocated by just 2 nt, were not apparently shifted by Chd1. To examine how interactions of Chd1 may differ for the $m=3$ bp versus other gapped nucleosomes, we used a single-cysteine Chd1 variant (N650C) that enables cross-linking between the ATPase motor and nucleosomal DNA. As shown in Supplementary Figure 8, cross-linking from Chd1(N650C) was stronger to the $m=3$ bp gapped nucleosome (lane 4). While this cross-linking suggests some differences in how the Chd1 ATPase motor may interact at SHL2, we cannot offer any more specific rationale for why Chd1 is unable to mobilize nucleosomes when the gap is too close to the ATPase binding site and consider this to be a negative result.

Given the resolution limit of 1-2 bp of the cross-linking technique, we refrained from additionally testing an $m=4$ bp gapped nucleosome. Based on these considerations, we

conclude that we cannot provide a more precise estimate than the range of 1-3 bp for the amount of DNA transiently buffered by the nucleosome.

Related minor point:

The authors show with smFRET analyses that a $m=0$ substrate (2nt gap at SHL2) leads to aberrant changes in FRET due solely to Chd1 binding. This seems very odd and the authors don't really follow this up. I was surprised that they did not try the $m=8$ substrate by smFRET or try the $m=0$ substrate with the crosslinking assay. It may be that an $m=3$ or $m=4$ substrate would yield interpretable results by smFRET. If so, this would boost the conclusions. In general it would be good to have overlap between these two different assays.

As the Reviewer correctly points out, we detected FRET changes due solely to Chd1 binding for the $m=0$ bp gapped construct. Following the Reviewer's suggestion, we have now additionally tested three more three-color labeled gapped nucleosomes that feature gaps at $m=3, 5,$ and 8 bp. All of these nucleosomes showed Chd1-dependent but ATP-independent FRET changes to some extent, indicating that gaps around this location result in structural changes not observed with non-gapped nucleosomes. These observations for these new substrates are reported in the revised manuscript in Supplementary Fig. 5.

While the internal DNA fluorophore labels allow the detection of ATP-dependent movements of ungapped nucleosomal DNA by Chd1, the backbone-incorporated dye labels (IDT DNA) result in an unpaired nucleotide opposite the dye. We speculate that this unpaired nucleotide at the the entry side, combined with a single-stranded DNA gap 28-36 bp away, may result in some unique structural perturbations that we detect by single-molecule FRET. Therefore, unfortunately, this ATP-independent FRET response upon addition of Chd1 prevents us from using both smFRET and cross-linking to monitor limited translocation of gapped nucleosomes. We point out our rationale for switching over from smFRET to cross-linking in the revised manuscript with this text [same as quoted above]:

“While internal dyes allow for ATP-dependent nucleosome sliding, we note that such backbone-incorporated labels result in an unpaired nucleotide opposite the dye. The presence of a 2 nt DNA gap at the entry SHL2, in combination with the unpaired nucleotide from the label, may give rise to structural perturbations of DNA that produce FRET changes independently of ATP hydrolysis. Further, FRET is sensitive to transient, non-canonical placements of the DNA on the nucleosome, which could obscure detection of ATP-dependent DNA translocation.”

Reviewer #3:

1. The authors determine that gapped DNA causes enough structural problems to prevent FRET studies (figure supplemental fig 6), but the authors trust the conclusions of the crosslinking work (figure supplemental figure 7 and fig 3). The authors should add some discussion about why they believe the structural problems caused by gapped DNA identified in the attempted FRET study do not impact the relevance of the conclusions from the crosslinking study.

We appreciate this point, also raised by the other Reviewers. As we responded above, single-stranded DNA gaps have been used by many groups to study nucleosome sliding, and while it is not clear exactly how they block DNA translocation, the ability of gaps to limit action of the ATPase motor has been a useful tool. Here, as we point out in the revised manuscript, we were concerned that the combination of the single-stranded DNA gap plus the unpaired DNA nucleotide opposite the dye may allow the nucleosomal DNA to behave differently than with the gap alone. This conclusion is supported by our results showing that Chd1 without nucleotide causes detectable smFRET changes for $m= 0, 3, 5$ and 8 bp gapped nucleosomes, indicating some perturbation upon Chd1 binding without hydrolysis (Supplementary Fig. 5). Therefore, we felt that examining gapped nucleosomes without the internal dyes would give a truer readout of how ungapped nucleosomes behave.

Additionally, cross-linking predominantly reports on DNA lying in the canonical DNA path around the histone octamer. Therefore, transient excursions away from the canonical nucleosome structure would reduce the overall amplitude of observed cross-links, but not alter the DNA translocation readout. In contrast, FRET is much more susceptible to such structural perturbations since non-canonical intermediate structures can yield complex FRET dynamics that would obscure the readout of how DNA changes position.

Moreover, by using cross-linking rather than single-molecule FRET, we can additionally monitor the position of DNA relative to the histone core at the nucleosomal dyad. While cross-linking does not have the time-resolution of single-molecule FRET, the ability to monitor entry, dyad, and exit sites provides a useful readout for the $m= 5$ and $m= 8$ bp gapped nucleosomes that can be repositioned by Chd1.

2. On page 7, at the beginning of the section “The onset of exit-side movement involves an additional ATP-binding event”, the authors state that “At subsaturating ATP concentrations...”. It would be helpful for non-experts to evaluate the relevance of their data if the authors cite and mention biochemical literature on the ATP affinity of the ATPase in Chd1. What are subsaturating or saturating concentrations for this enzyme?

We thank the Reviewer for bringing this to our attention. Several papers have shown that nucleosome sliding slows down with lower ATP (Blosser et al., 2009; Deindl et al., 2013), and we therefore added the following text to better put our data more into context:

“For Chd1, the K_M for ATP was previously determined to be 50-60 μ M (Patel et al., 2011). At subsaturating ATP concentrations, nucleotide binding limits the observed rate of nucleosome sliding (Blosser et al., 2009; Deindl et al., 2013). ”

3. On page 15, in single-molecule FRET section of methods, it is stated that Cy3 and Alexa750 signals are scaled to correct for differences in QY and detection efficiency. Some readers would be interested to know those scaling factors and they could be added to the methods.

In the revised manuscript, we now include details on the correction for differences in quantum yields and detection efficiencies as well as bleed-through and direct excitation (Supplementary Note 1). To further illustrate these corrections, we have included a new supplementary figure showing a direct comparison of raw and corrected time traces (Supplementary Fig. 9).

Reply to Reviewers' comments, round 2:

Reviewer #1 (Remarks to the Author):

The revised version of the manuscript "Direct observation of coordinated DNA movements on the nucleosome during chromatin remodelling" is considerably improved and my comments have been carefully addressed. For some of the data, the authors added automated detection of lag times even though the answer "visual inspection" would have been sufficient for me.

Minor:

In their response, the authors suggest that the fluorescently labelled nucleotide in the DNA remains unpaired. That surprises me as the base pairing should not be abolished using an internal amino modifier C6 dT as long as the other strand contains a complementary dA? Or is there no complementary dA present? If this is the case, what was the authors rational here?

We thank the Reviewer for their supportive and thoughtful comments. In response to the minor point: For our labeling scheme, the internal Cy3 dye replaces a DNA base, and therefore the presence of this dye may disrupt normal base stacking (i.e. the dye is opposite an unpaired base). We previously used iCy3 successfully for monitoring nucleosome sliding, which is why we chose it for this study. In the previous response to Reviewers' concerns, we incorrectly stated that the Alexa750 fluorophore also replaces a DNA base; this modification, however, extends from a modified dT. We apologize for this oversight, and have clarified these properties of both dyes in the methods section. However, despite maintaining a T-A base pair at the entry Alexa750 dye, we still believe that it is a combination of the internal dyes with the DNA gap that is responsible for the observed ATP-independent FRET changes. As we stated in the previous response to Reviewers' comments, histone cross-linking is less sensitive to transient structural perturbations than smFRET, and therefore more suitable for following DNA movements on gapped nucleosomes.

Reviewer #2 (Remarks to the Author):

The authors have fully addressed my previous concerns. In particular, I appreciate that they have added a number of new experiments with the gapped nucleosomal substrates - not a trivial addition! This is an excellent manuscript.

We thank the Reviewer for their insightful and supportive comments.

Reviewer #3 (Remarks to the Author):

The revised version of the manuscript is very strong. The authors have addressed the comments of the reviewers. They have also added new experiments with additional substrates and an additional remodeler protein. I have no further concerns. The manuscript provides a strong addition to the field.

We appreciate the Reviewer's valuable input and enthusiasm for our manuscript.